# Barriers and facilitators of loaded self-managed exercises and physical activity in people with patellofemoral pain: understanding the feasibility of delivering a multicentred randomised controlled trial, a UK qualitative study

Benjamin E Smith,[1,2] Fiona Moffatt,[3] Paul Hendrick,[3] Marcus Bateman,[1] James Selfe,[4] Michael Skovdal Rathleff,[5,6] Toby O Smith,[7] Phillipa Logan[2]

For numbered affiliations see end of article.

**Correspondence to**
Benjamin E Smith; benjamin.smith3@nhs.net

## ABSTRACT

**Objectives** There is an emergent body of evidence supporting exercise therapy and physical activity in the management of musculoskeletal pain. The purpose of this study was to explore potential barriers and facilitators with patients and physiotherapists with patellofemoral pain involved in a feasibility randomised controlled trial (RCT) study. The trial investigated a loaded self-managed exercise intervention, which included education and advice on physical activity versus usual physiotherapy as the control.

**Design** Qualitative study, embedded within a mixed-methods design, using semi-structured interviews.

**Setting** A UK National Health Service physiotherapy clinic in a large teaching hospital.

**Participants** Purposively sampled 20 participants within a feasibility RCT study; 10 patients with a diagnosis of patellofemoral pain, aged between 18 and 40 years, and 10 physiotherapists delivering the interventions.

**Results** In respect to barriers and facilitators, the five overlapping themes that emerged from the data were: (1) locus of control; (2) belief and attitude to pain; (3) treatment expectations and preference; (4) participants' engagement with the loaded self-managed exercises and (5) physiotherapists' clinical development. Locus of control was one overarching theme that was evident throughout. Contrary to popular concerns relating to painful exercises, all participants in the intervention group reported positive engagement. Both physiotherapists and patients, in the intervention group, viewed the single exercise approach in a positive manner. Participants within the intervention group described narratives demonstrating self-efficacy, with greater internal locus of control compared with those who received usual physiotherapy, particularly in relation to physical activity.

**Conclusions** Implementation, delivery and evaluation of the intervention in clinical settings may be challenging, but feasible with the appropriate training for physiotherapists. Participants' improvements in pain and function may have been mediated, in some part, by greater self-efficacy and locus of control.

### Strengths and limitations of this study

► This paper identified, through interviews, key barriers and facilitators to implementation of a loaded self-managed exercise programme, with education and advice on physical activity.
► Two authors independently coded all transcripts, and a clear, transparent and reproducible methodological approach was used in the analysis.
► The main limitations of this study were the difficulty in interviewing patients lost to follow-up (from both groups) and finding patients classed as 'non-responders' in the loaded self-managed group.
► The study population comprised a single clinical setting, where the researcher was also a clinician.

**Trial registration number** ISRCTN35272486; Pre-results.

## INTRODUCTION

Patellofemoral pain (PFP) is one of the most common forms of knee pain in adults under the age of 40 years, with an estimated prevalence of 23% in the general population.[1] Many individuals with PFP develop associated pain-related fear, such as fear-avoidance and catastrophising thoughts in relation to their knee pain.[2–4]

This research was undertaken within a framework of mixed-methods, embedded within a feasibility study comparing a loaded self-managed exercise protocol with usual physiotherapy for people with PFP.[5] The loaded self-managed exercise programme is a novel intervention based on pain science (where a single exercise is designed to load and temporarily aggravate patients' symptoms), self-management strategies and

improvements in physical activity levels.[5] Usual physiotherapy can be described as a mixed packaged (multi-model) approach of 'trial-and-error' exercises, patellar taping and bracing, and foot orthoses. It is typically aimed at reducing the load on the patella, with avoidance of painful exercise.[6 7]

The loaded self-managed exercise programme does not align with current UK physiotherapists' preferred treatment approach for PFP.[7] First, protocols that use loaded exercises are typically painful to perform,[5] thought a strong predictor of poor adherence.[8] Second, pain education and increasing physical activity require a certain level of self-management and personal responsibility on the part of the patient, also strong predictors of poor exercise adherence.[8] And third, a key aspect of the loaded self-managed exercise programme is the single exercise method, which physiotherapists and patients historically viewed with a degree of scepticism, when used in treating shoulder pain.[9 10]

Therefore, this qualitative investigation aimed to explore potential barriers and facilitators to implementation of the intervention with participants with PFP involved in a feasibility randomised controlled trial (RCT),[5] with acknowledgement that qualitative inquiry can provide insights that may lead to development of ideas and hypothesis generation.

## METHOD

A qualitative study was conducted embedded within a mixed-methods feasibility study. To avoid cross-contamination between the two groups, the intervention group was treated by different qualified physiotherapists, who received the intervention training package, to the usual physiotherapy group. To fully explore the aims of this study, patients and physiotherapists receiving and delivering both the intervention and usual physiotherapy were interviewed.[5] The framework approach was the most appropriate method for inquiry, as the objectives of the investigation were set *a priori*.[11]

This study has been reported in line with the COnsolidated criteria for REporting Qualitative research checklist.[12]

This study did not set out to prove or disprove a hypothesis, it set out to generate new data from which an understanding of barriers and facilitators to the intervention and study design might be developed. The authors took an epistemological position described as 'contextualist' by Braun and Clarke that sits central on the spectrum of realism and constructivism.[13] It recognises the experience at an individual level, while considering the wider context within a sociocultural perspective. Through this, the beliefs and perceptions of a person, with any meanings attached, can be explored, while considering social and cultural factors. This position has previously been discussed in detail in relation to this mixed-methods study.[2]

## PARTICIPANTS

A purposive sample of 10 patients with PFP were recruited from the 60 patients who were recruited to a feasibility study, this included patients in the intervention group and those receiving usual physiotherapy. International consensus has defined PFP symptoms as typically developing insidiously with retropatellar pain or diffuse peripatellar pain, aggravated by activities that 'load the joint', such as climbing and descending stairs, squatting, running or jumping.[14] Based on similar studies, we anticipated this sample size would be sufficient to reach data saturation.[9 10] Patients were selected based on representation of a spectrum of population in terms of: intervention delivered (both the intervention, and usual physiotherapy), age, gender, return of outcome forms and clinical outcome, as determined by a global rating of change at follow-up measured on a 7-point Likert scale ranging from 'completely recovered' to 'worse than ever'.[5] Clinical responders were defined as 'completely recovered' or 'strongly recovered'.[5] Attempts were made to interview those lost to follow-up and non-responders in both groups.

Initial recruitment to the feasibility study included gaining written consent for taking part in future qualitative investigations with consent to audio-recording and to publication of anonymised quotations. Participants were initially followed up by a telephone call. If they agreed, a convenient time was arranged to complete an interview. Participants were given the opportunity to discuss any concerns before the interviews started.

Ten physiotherapists were purposively sampled, this included those delivering the intervention and those delivering usual physiotherapy. Based on similar studies, we anticipated this sample size would be sufficient to reach data saturation.[9 10] Again, physiotherapists were selected based on characteristic to represent a spectrum population in terms of: intervention delivered, age, sex and length of time qualified. The physiotherapists initially agreed to take part in the research when briefed during the study intervention training sessions. They were subsequently approached about the qualitative component of the study via team meetings. Participants were given the opportunity to read the participant information sheet and to ask any questions before the consent form was signed.

## RECRUITMENT

All participants were interviewed at a convenient time in the hospital-based physiotherapy department. The researcher (BES) introduced himself as a physiotherapist working in that department, and as a researcher conducting a PhD. The researcher explained the aims of the study. Verbal consent was taken to start recording.

## DATA COLLECTION

Semi-structured interviews were designed by the researchers (BES and FM) using topic guidelines with

prompts to explore barriers and facilitators to taking part in a loaded self-managed exercise intervention. Patients from both treatment groups were asked about response to treatment, belief and attitude to pain, belief and attitude to physical activity, treatment expectations and protocol parameters. Only those in the intervention group were asked about their engagement with the loaded self-managed intervention. All physiotherapists were asked about their usual practice, personal development, belief and attitude to pain, belief and attitude to physical activity and protocol parameters. Only those delivering the intervention were asked about their engagement with the loaded self-managed intervention, including the training package. The interviews ranged from 5 to 21 min (mean time 11 min) in duration.

The interview guide was not piloted, however the researcher maintained a reflective journal, noting down initial thoughts and ideas after each interview.[15] This identified that the first two interviews raised matters relating to responsibility and locus of control around return to physical activity. This was incorporated into subsequent interview schedules for both patients and physiotherapists.

## DATA ANALYSIS

All audio files were collected and transcribed verbatim.

The data were analysed using a thematic Framework Method,[11] which was the most appropriate method for inquiry, as the objectives of the investigation were set a priori.[11] Furthermore, data analysis can be conducted systematically, allowing the data to be explored in depth while simultaneously maintaining an effective and transparent audit trail.[11] During transcription, initial thoughts and ideas were noted in the reflective journal. Audio files were listened to several times to check for accuracy, and transcriptions were read and re-read a number of times; these data familiarisation further informed the development of a thematic framework. Following familiarisation, both authors agreed on the initial thematic framework. Data coding then identified and coded pertinent features of the data giving equal priority over the whole dataset. These steps were independently conducted by two researchers (BES and FM) who met to compare codes. This formed a working analytical framework on which the data were examined. The transcripts were then indexed using the categories and codes on the working framework. During this process, the data were organised according to the defined thematic framework. Charting was then used to summarise and display the data by category and theme for each transcript.[11 16] Indexing was initiated by one researcher (BES), prior to charting, and subsequently developed and verified by a second researcher (FM).

Data were organised and analysed using QSR International's NVivo V.11. After 10 interviews per group, it was determined by the researchers that data saturation had occurred as no new thoughts or concepts were generated in the later interviews.

### Table 1  Characteristics of patients

| Participant number | Gender | Intervention received | Clinical responder |
|---|---|---|---|
| P1 | M | Intervention | Responder |
| P2 | M | Usual physiotherapy | Non-responder |
| P3 | F | Usual physiotherapy | Non-responder |
| P4 | F | Usual physiotherapy | Responder |
| P5 | F | Intervention | Responder |
| P6 | F | Usual physiotherapy | Non-responder |
| P7 | F | Usual physiotherapy | Responder |
| P8 | F | Intervention | Non-responder |
| P9 | M | Intervention | Responder |
| P10 | F | Intervention | Responder |

F, female; M, male.

### Patient and public involvement

This research project has been driven by the views of people suffering from PFP. Patients were consulted for their views, including patient members of the Steering Group Committee. Thoughts and preferences to current programmes of therapy and treatment were requested, and these views have been incorporated into the planning, design, application and dissemination of this study.

### RESULTS

The 10 patients included 3 men and 7 women, aged between 26 and 37 years (mean: 30.6 years), with a diagnosis of PFP for a mean duration of 25 months (range: 3 months to 10 years). The 10 physiotherapists included 2 men and 8 women, aged between 24 and 58 years (mean age 39.4 years), with a mean of 16 years qualified (range: 3–37 years). Full patient and physiotherapist characteristics are presented in table 1 and table 2, respectively.

In respect to barriers and facilitators, the five major overlapping themes that emerged from the data were: (1) locus of control; (2) belief and attitude to pain; (3)

### Table 2  Characteristics of physiotherapists

| Therapist number | Gender | Intervention delivered |
|---|---|---|
| T1 | F | Usual physiotherapy |
| T2 | F | Intervention |
| T3 | M | Intervention |
| T4 | F | Intervention |
| T5 | F | Usual physiotherapy |
| T6 | F | Usual physiotherapy |
| T7 | F | Intervention |
| T8 | M | Intervention |
| T9 | F | Usual physiotherapy |
| T10 | F | Usual physiotherapy |

F, female; M, male.

treatment expectations and preference; (4) participants' engagement with the loaded self-managed exercises and (5) physiotherapists' clinical development. Locus of control was one overarching theme that was evident throughout. The findings are presented in relation to existing literature.

### Theme 1: locus of control

Locus of control is a psychological construct about the degree people believe they have control over their actions and outcomes.[17] A key feature of the intervention being evaluated in the RCT is the self-dosing of exercise, based on the symptomatic response, and the self-managed approach to physical activity. This could be conceptualised as internalising locus of control with the patient, and is thought to predict treatment compliance, acting as a barrier or facilitator to implementation.[8] Patients within the intervention group described narratives that could be conceptualised as greater internal locus of control, compared with patients in the usual physiotherapy group.

> R: And how did you feel about being in charge of that (the exercise)?
>
> P8: Yeah. I think it was empowering in a way. (Loaded self-managed)

Early interviews raised matters relating to whose authority it was to give the 'permission' to return to, or increase, physical activity; including when and how this should be done. Again, clear differences between usual physiotherapy and the intervention could be seen, particularly in relation to physiotherapists' management approach to physical activity.

> Ultimately up to the patient really. They should feel in charge of what they do. They need to have control of the situation. If they're just waiting for somebody else to dictate that, then they haven't got very good control. But they might need some encouragement or reassurance that it's okay to actually, if you want to get back to these activities you can. You don't need to ask me permission really. (T2—loaded self-managed)
>
> I would usually kind of bat it back to them and say, 'Well, what do you think you can do?' And using the same principles as with the exercises, if you're getting some discomfort at the time, it doesn't mean to say you then stop. And just see how it is afterwards, and then modify how much you're doing in response to how much pain you're experiencing afterwards. (T4—loaded self-managed)

Contrasting the push for an internal locus of control with the intervention was a narrative discussed by some patients receiving usual physiotherapy. For example, participant 4 had indicated she was 'strongly recovered', had minimal pain and had returned to almost all of her usual activity. However, she had not returned to the gym yet, and had booked a follow-up appointment with the treating physiotherapist for after the interviews where she hoped to receive the 'go-ahead' to return.

And this patient narrative was reinforced by the treating physiotherapists' understanding of their role:

> I'd assess them functionally. So you kind of break down that hobby or that activity into sections. So if it's a sport, look at part of it… and if you can't do two or three of them, it's not just your knee that's letting you down. Generally, you're not quite ready for that. (T10—usual physiotherapy)

A few of the physiotherapists within the usual physiotherapy group viewed their role more of a partnership with the patient, where decisions about return to activity were agreed mutually.

> Well, it'd be a mutual thing. A lot of them weren't sporty, but they would ask and we discussed the suitability. (T5—usual physiotherapy)

Locus of control is inter-related to the psychological construct of self-efficacy, where it relates to the power of thinking in achieving treatment outcomes.[18] The loaded self-managed exercise programme is designed around optimisation of self-management and self-efficacy. For example, the progressive hierarchy of the exercise demonstrates and provides evidence to the patient that they are systematically approaching their clinical and personal goals.[19] Some patients within the intervention group expressed views that could be contextualised as self-efficacious in line with this hierarchy.

> That sense of just you know how much progress you made. A week ago you did 20, and now you did 30 or 40. (P9—loaded self-managed)
>
> When I hit the target and I then thought, 'Oh, I can actually do a few more', and it's comfortable to do, I did do that. (P5—loaded self-managed)

### Theme 2: treatment expectations and preference

Previous qualitative work has identified unmet treatment expectation as a potential barrier to treatment adherence,[20 21] therefore all patients were asked to reflect on their expectations, with physiotherapists invited to discuss their usual practice. The predominant patient expectation was that they would receive some form of exercise programme from their physiotherapy, and that this would probably involve some level of pain.

A small number of patients discussed an expectation of hands-on passive treatment.

> I was more expecting sort of a hands-on approach, more like physio massage when I came. (P8—loaded self-managed)

Furthermore, in keeping with themes found in other PFP qualitative work,[2] several patients established a clear wish for questions to be answered, in relation to causative factors around their pain:

For me, I wanted answers on why my knee was painful. Because I think, going back 10 years ago, when I first went to my doctor's, I was told it was ligament damage. And it didn't clear up, and when I went back, it was like, 'Well, the waiting list for physio is so long, by the time you get there, you'll be recovered'. And then, when I went back again, it was like, 'Well, you're too young to have steroid injections'. And then, I just always felt I was like, in a sense, sent packing without any answers. And then, I wanted some answers as to why it's hurting so I could understand it. (P10—loaded self-managed group)

Previous qualitative work in patients with PFP found a dominant negative view of physiotherapy,[2] with one patient similarly expressing an initial negative view of seeing a physiotherapist.

The physio—I don't know, I was a bit sceptical, to be honest. But yeah, it has given me the result I wanted. (P10—loaded self-managed)

All physiotherapists reported that their current practice and preference for treating PFP included an exercise programme. However, in contrast to the majority of UK physiotherapists,[7] they all reported an expectation that exercises would be performed with a degree of pain. Although there remained a large amount of heterogeneity in terms of language choice, and what parameters were used, when discussing optimal exercise dosage with patients.

But if you think about a VAS or something like that … probably you wouldn't want your pain to be greater than maybe a 3 or a 4 out of 10. (T1—usual physiotherapy)

Quite often I tell people to do reps to kind of fatigue, but not to pain. So people are getting a bit of a niggle, if they can manage it, and they can bring the pain level back down quite quickly afterwards. So if they can do exercises, it aggravates it, but within about a half an hour symptoms have settled, then that's fine. (T10—usual physiotherapy)

Dissonance between the single exercise approach used in the intervention and treating physiotherapists' preference was evident. The single exercise approach was not favoured by any of the physiotherapists interviewed:

I think possibly the intervention was simpler to do in the fact that it was geared, sort of guided around one exercise. And probably, what I would have done before is perhaps give more exercises and chop and change them maybe a bit more frequently. (T7—loaded self-managed)

Additionally, some physiotherapists were very prescriptive with their exercise dosage.

Initially I might start with them with 15 repetitions and work to three sets, 2 min break in between. (T9—usual physiotherapy)

Again, in contrast to the majority of UK physiotherapists,[7] and similarly to the experimental intervention, many of the physiotherapists interviewed in this study (from both groups) would try to encourage the patient to self-dose their exercise:

I'm a little less strict on sets and reps. I'm more do what you feel you can. If you're happier, push on a little bit more. (T3—loaded self-managed)

As identified above, most patients were content with the anticipation that exercises would be painful, and indeed this matched current clinical practice with the physiotherapists interviewed, despite not aligning with UK wide current practice.[7] Where departmental practice did align itself more with UK practice, was with regard to the number of exercises prescribed, in clear contrast to the single exercise approach with the intervention.

### Theme 3: belief and attitude to pain

Interlinked to the all themes, particularly locus of control were patients' and physiotherapists' beliefs and attitudes to pain. There is a growing body of evidence suggesting that health practitioners with a biomedical orientation to pain are more likely to advise patients to limit their physical activity due to pain[22–24]; and consequently may induce fear-avoidant behaviours onto their patients,[24 25] acting as a clear barrier to implementation. There were examples in the usual physiotherapy group of biomedical models of diagnosis and management with misconceptions of 'tissue damage':

She (the physiotherapist) gave me exercises to do. I've always been keen on the gym. I go to the gym. I was a doing a lot of the stuff she's asking me to do, anyway. Or it's probably more about my technique. I was maybe not doing it as well as I could have done. So I fell back. …So she referred me for scans on both knees—well, referred me back to my doctor. My doctor referred me to an orthopaedist. They referred me for a scan on both knees. The MRI scan showed this knee's absolutely fine—which it's not. (P3—usual physiotherapy)

R: So if they're not achieving that, would you advise them not to run then?

P10: Probably. Yes. I'd probably have a look at them, and if they were really antalgic on their gait, then yeah, tell them not to bother, to work on their weaknesses, and then reassess it a bit later down the line. Because otherwise, they might just end up making their knee 10 times worse because they're running on a weakened, less controlled knee. (Usual physiotherapy)

Of interest is that the physiotherapist delivering the usual physiotherapy, as described in theme 2, did describe treatment preference not fully aligned with the majority of UK physiotherapists,[7] and the best practice guidelines,[6] in as much as they expressed a belief that pain is

acceptable during exercise. Certainly, this did identify some fidelity and contamination concerns with regard to usual physiotherapy:

> I think it was sometimes a bit hard to stick to usual physio, because we still keep reading. We try to keep up with what's happening… So it's just a bit of reading and then I change 'usual physio', it keeps developing as you work. (T9—usual physiotherapy)

Yet despite this, there was marked differences in the patients' and physiotherapists' beliefs and attitudes to pain in the intervention group, compared with usual physiotherapy, demonstrating some reconceptualisation of pain. This suggests the training programme did improve contemporary knowledge of pain science.

> Yeah, the pain wasn't excruciating or anything. At no point did I think, 'I can't keep doing this'. It was a fairly normal level, I'd say. It wasn't anything that would make me come back, and say, 'I'm worried that I'm doing something wrong', or anything like that. It was fairly normal. I wouldn't say it was too bad. (P1—loaded self-managed)

> P7: The physiotherapist said to go ahead and run if it wasn't going to do any damage. Yes, if it's painful, stop. (Usual physiotherapy)

> My own thoughts have been, I think, changed definitely with this intervention. I think exercise is—I've always said to patients that if it's painful, they can still carry on. But again, like I said, I gave that arbitrary figure. If it goes above this, then maybe taper down… But actually, maybe educating them and telling them, 'Pain isn't an indicator of damage. You can push through into it a little bit, but it just has to be something that you're comfortable with'. And I think the thing that changed with me saying that to patients was I am not the one that's going to dictate that. You're the one has to go through this. (T3—loaded self-managed)

There was one example of mixed messages from the patient, with regard to acceptable and appropriate levels of pain during exercise and physical activity. This may suggest the heterogeneity in physiotherapy advice, as previously discussed in the second theme with physiotherapists, may have a negative effect with increasing levels of uncertainty. This is in keeping with previous research suggesting an iatrogenic effect with physiotherapy treatment for PFP relating to diagnosis uncertainty and fear-avoidance behaviour.[2]

> He (the physiotherapist) recommend that I didn't run, which is probably the only thing I don't do now. I think it was the impact. Like, my knee with my cartilage. That's why he didn't recommend it at that point. (P10—loaded self-managed)

## Theme 4: participants' engagement with the loaded self-managed exercises

Only patients and physiotherapists receiving or delivering the intervention were asked to discuss their thoughts about it. Both patients and physiotherapists reported several different ways in which they interacted and connected with the intervention. First, the intervention laid the foundation of reconceptualisation of pain-related fear where the physiotherapist spent a period of time educating the patient about pain mechanisms.[5] Descriptions of tissue-based pathology models of pain, for example, patellar maltracking, or limb malalignment were actively discouraged and challenged by the physiotherapist. The aim was for the patient to gain an evidence-based understanding of dysfunctional central nociceptive processing as an explanation of chronic and persistent pain and the role and impact of fear.

> Once you'd explained—all the key is in the explanation about pain and how pain works and explaining why they're doing it from that. And in fact, sort of the particular girl I'm thinking about, she'd stopped going downstairs because of the pain. When I reviewed her last time, she said, 'Well, I haven't been avoiding the stairs' (with no increase in pain levels). So it's good stuff. (T7—loaded self-managed)

Other critical aspects of the intervention discussed by the participants were the self-dosage of the exercise, based on the symptomatic response, rather than being prescribed by the physiotherapist. These aspects were all discussed positively, with no negative features identified.

> I think for me I've got results a lot quicker, so because I was kind of going through the pain with all that. And I definitely stuck with the exercise more, because when I first started with one exercise I might get a bit bored. But I've definitely stuck to it more. (P9—loaded self-managed)

The simplicity of a single exercise approach was discussed by all the interviewees, predominantly in a positive manner.

> So I think it's quite simple, so if I do ever get—the problem starts to occur again, it's no real problem to just start. (P1—loaded self-managed)

However, one physiotherapist admitted to being initially sceptical that one exercise would be enough.

> And using that single exercise as that treatment. So in terms of my thoughts before, would that be enough for my patients? And the ones I've seen, have seemingly done well with just one exercise, rather than having four or five different exercises to do. (T3—loaded self-managed)

The key feature of patients self-dosing their exercise, based on the symptomatic response, is an understanding of when and how to progress or regress the exercise. Patients recognised the role of 'trial and error' in this

process, and the relevance of the pain education prior to the exercise programme being implemented.

I do remember, initially, there being kind of a week or two, maybe, where I was kind of finding kind of the right amount (of the exercise to do). (P9—loaded self-managed)

I think what you tend to do as physios, we very often tend to be quite prescriptive. And patients do ask that. They want to know how many they should do, how many times a day, whereas this is actually giving them much more their own power of making them decide what they're going to do. So actually, hopefully, then they're going to carry on with it in the future. (T7—loaded self-managed)

Interlinked to self-dosing was the expected pain flare-ups, when patients overdosed their exercise or physical activity. The physiotherapists' training programme at the start of the feasibility study covered this topic, with physiotherapists aiming to discuss self-management approaches at preventing and dealing with flare-ups. Despite this, flare-ups remained common place, and were a cause of concern for several patients; suggesting this topic needs additional emphasis in any future training programme.

R: Did it worry you when you had those flare-ups?

P1: Yeah. There were kind of back-of-your-head thoughts, like, 'What if this time I have done it a bit too far? If it lasts a bit longer, am I going to have to go back in case I've damaged it a bit?' or anything like that. But most of the time, again, was 2 days tops. So I did have kind of a little niggling worry, but nothing to kind of cause me to do anything or anything like that. (Loaded self-managed)

Both patients and physiotherapists were asked to reflect on the intervention and their clinical response. For patients, quantitatively, the global rating of change at follow-up (measured on a 7-point Likert scale ranging from 'completely recovered' to 'worse than ever') was used to identify responders and non-responders. The scale was dichotomised so that responders were defined as 'completely recovered' or 'strongly recovered',[5] and patients were purposively sampled to ensure that responders and non-responders were included. However, one patient (participant 8) who received the intervention identified quantitatively as a non-responder. However, qualitatively all five patient participants interviewed from the experimental arm reported improvement and satisfaction with the loaded self-managed intervention.

Yeah. I'm playing football again. Yeah. I'm just kind of—sometimes I can tell I've got a little bit of tension there. But I'm not getting pain. It's not stopping me doing nothing at all. So yeah. (P9—loaded self-managed)

And this corresponded from the feedback from the treating physiotherapists, with all physiotherapists reporting favourable outcomes with the intervention.

The main emphasis of patients' and physiotherapists' narrative was the simplicity of the exercise, the loaded element of the exercise and the self-dosage of the exercise.

### Theme 5: physiotherapists' development

It is thought that difficulties accessing and understanding research, and professional isolation may act as barriers to implementation of research into practice.[26] Therefore, treating physiotherapists, in both the usual physiotherapy and intervention groups, were asked to reflect on their clinical development. Particularly on beliefs around pain and exercise, and how they have developed their management approach to PFP. There was a common theme among all physiotherapists of clinical development over the preceding few years, with concomitant changes within their management approaches. This reflection attributed some of this development, in part, to working within a department where clinical trials were being undertaken, with exposure to contemporary thinking and practice.

I don't think I ever would have said to people, 'Don't push into any pain'. I think over the years I've probably got—as research projects and things we've done where we're kind of talking more about it being okay to push into pain, I've got more relaxed with it… I think maybe as a junior I might have done, to be honest. So probably when I did my first rotation, I might have been saying more, 'Very, very low', or, 'It needs to be virtually pain free'. But as the years have gone on, probably got more and more relaxed with saying it's okay, on the back of, I suppose, of the things that have happened in our department and changes in practice generally. (T1—usual physiotherapy)

I think from when I first started practice, it would have been different. So when I first started, I would often tape the knee, or if they came back and said that it was painful, I asked them to kind of back off. Almost think about off-loading the knee if it was painful. So trying to reduce activity if it was sore. And then I think just as I became more experienced and read more about that type of thing, I got more confident in not using adjunct and trying to use loaded exercise and reassurance about pain. So I think it fits more with my current practice, and I don't think it was that different. Obviously, I do a lot of pain education with back patients, so I think that was quite easily transferable. (T8—loaded self-managed)

Department culture has been identified in previous qualitative work as a facilitator or barrier to change, over and above research evidence and clinical guidelines,[27 28] and the physiotherapists within this study also reflected on department culture as a driver of practice.

I guess in this department we're quite used to doing that sort of intervention for these patients, so it wasn't

particularly ground-breaking to me, in a nice way (laughter). It's your (the researcher's) fault. (T2—loaded self-managed)

Oh, it is working in a different environment as well. So when I was in ** I was most of the time by myself in a GP clinic. And you don't get a lot of interaction. That influence, when you actually have a bigger (department). We talk about loading as well. So we talk about Achilles or tendons and we just keep talking about how everything changes and you just do your own research and you think, 'Okay'. How to make it better. (T9—usual physiotherapy)

Two physiotherapists discussed how being part of the research challenged their current practice and resulted in clinical development to both patients with and without PFP. One physiotherapist conferred how the training package and personal reflection of treating study patients challenged him; the second from sparking an interest in research.

I think if you tell them, 'Actually, how do you feel about it. You're in control', gives them the onus to take what they do. That's definitely changed massively. And I kind of do that with other patients now as well, not just the knee patients. I'm a little less strict on sets and reps. I'm more do what you feel you can. If you're happier, push on a little bit more. (T3—loaded self-managed)

## DISCUSSION
### Main findings

In respect to barriers and facilitators, the five major overlapping themes that emerged from the data were: (1) locus of control; (2) belief and attitude to pain; (3) treatment expectations and preference; (4) participants' engagement with the loaded self-managed exercises and (5) physiotherapists' clinical development. Locus of control was one overarching theme that was evident throughout.

The aim of this qualitative study was to identify barriers and facilitators to the implementation of a loaded self-management exercise programme, which included education and advice on physical activity. Contrary to popular concerns relating to adherence of painful exercises,[7 8 29] all patients in the intervention group reported positive engagement. However, flare-ups from over dosing occasionally happened, with some patients expressing concern over reoccurring thoughts of 'tissue damage'; this may be relevant to all patients receiving an exercise programme. This topic needs additional emphasis in any future training programme delivered to the physiotherapists, for example, with an addition of a dedicated objective in the training package, or via case-study workshops. Previous research has identified physiotherapists' negative beliefs around pain and exercise as a potential barrier to loaded exercises,[10] but this was not apparent

with the physiotherapists from both groups interviewed in this study.

A key aspect of the loaded self-managed exercise programme is the single exercise method. Previous research with a similar approach in patients with shoulder pain identified this as a potential barrier to implementation, with physiotherapists and patients viewing this with a degree of uncertainty and scepticism.[9 10] However, contrary to this research, and despite not aligning with the physiotherapists' usual practice, both physiotherapists and patients generally viewed the single exercise approach in a positive manner. Furthermore, there was a general underlying acknowledgement of the key benefits of a single exercise approach, from both patients and physiotherapists, in terms of a time-saving approach aimed at optimising adherence, and improved dosage monitoring.

Locus of control is thought to predict health-related behaviours and physical activity,[30] with an important concept that it may predict healthcare utilisation.[31] Locus of control and the psychological construct of self-efficacy has overlapping meaning, where it relates to the power of thinking in achieving treatment outcomes.[18] The loaded self-managed exercise programme is designed around optimisation of self-management and self-efficacy. For example, the progressive hierarchy of exercises[19]; self-dosage of the exercise; mastery of a single exercise approach and self-management strategies for physical activity engagement, providing the foundations for self-management of flare-ups, are intended to reduce the need for direct physiotherapy intervention. It has been shown that the lack of belief in one's own ability to manage and function despite pain is a significant predictor of which individuals with pain become disabled or depressed, with regression analysis showing that self-efficacy mediates the relationship between pain and disability.[32] Within the context of this study, patients in the intervention group described narratives that could be conceptualised as self-efficacious with greater internal locus of control, compared with patients in the usual physiotherapy group. This could be seen particularly in relation to return to physical activity; belief and attitude to pain; engagement of the intervention with self-dosage of the therapeutic exercise and self-management.

### Clinical and research implications

Previous qualitative work has suggested that department culture is a key driver or barrier to change.[27 28] Indeed, there were clear examples of department culture within this study directly driving recent changes in physiotherapists' clinical practice. This matched previous physiotherapy qualitative work that has identified reflexion of practice and implementation of change, perhaps expeditiously, in physiotherapists who are directly engaged in research.[10] With recent research demonstrating that research active hospitals have better patient outcomes,[33] this may be considered a good thing. However, the results of this qualitative study suggest that in departments which

are actively engaged with research, clinical practice may be driven by members of the research team, in lieu of definitive research results or clinical guidelines. Considering the lead researcher works in the department where the interviews were conducted, and may in part drive department culture, implementation of the intervention in other departments may be more complicated.

Implementation fidelity refers to the degree by which the delivery of an intervention adheres to the protocol and description.[33] Physiotherapists delivering usual physiotherapy differed from the UK's usual practice, and best practice guidelines, largely with regard to the advice given on tolerable levels of pain during exercise and physical activity, and how the number and repetitions of the exercises are prescribed.[6 7] Cluster randomisation, where intervention and control participants are located at different recruitment sites, is one way of overcoming what is referred to as 'contamination'.[34]

This research demonstrates that even though physiotherapists have certain expectations around management and exercise prescription, their approach was adaptable to the intervention with only two, 2-hour training sessions; enabling patients to self-manage and make sensible decisions about their own treatment and return to physical activity. The results of this study establish a skill set needed to deliver the intervention, including: complex musculoskeletal assessment; anatomy; tissue healing and remodelling; pain biology; peripheral and central sensitisation; psychological and social factors that might affect pain perception; self-management strategies and education skills. Currently, in the UK, these skills form part of the degree training programme for physiotherapy, further supplemented by the research training package.

### Study limitations and strengths

Two authors independently coded all transcripts, and used a clear, transparent and reproducible methodological approach to data analysis. The author's clinical and research experience lie within the biopsychosocial framework of musculoskeletal pain. It is worth noting that the interviewer made it explicit to the participants that he was a physiotherapist working in the department conducting the research.

Despite efforts to the contrary, the main limitations of this study were the difficulty in interviewing patients lost to follow-up (from both treatment groups) and those classed as non-responders in the experimental intervention group. Four patients were contacted who failed to return any outcome measures, and initially agreed to be interviewed; unfortunately, they failed to attend.

The study population comprised a single clinical setting, where the researcher was also a clinician and where clinical trials are often undertaken; it is unknown how transferable the intervention is without the relevant physiotherapy training package.

It is possible that the patient sample may differ from other samples within the UK, and how representative these findings are to other populations with PFP is unknown.

### CONCLUSION

This qualitative paper has identified some of the barriers and facilitators with participants (physiotherapists and patients) with the delivery of a loaded self-managed exercise programme, with education and advice on physical activity.

From the patients' perspective, facilitators to engagement included effective education around: self-management on exercise dosage; physical activity and flare-ups. This facilitation may have been mediated, in some part, to enhancements of self-efficacy and internalised locus of control. From the physiotherapists' perspective, these results highlight the importance of 'control' and self-management during their assessment and management of patients with PFP.

In the context of the UK's usual management approach for PFP, which showed that a large proportion of practising physiotherapists would advise a patient to cease exercise or physical activity if they experience pain, implementation into general clinical practice may be challenging, but, ultimately, feasible.

**Author affiliations**
[1]University Hospitals of Derby and Burton NHS Foundation Trust, Derby, UK
[2]Division of Rehabilitation and Ageing, School of Medicine, University of Nottingham, Nottingham, UK
[3]Division of Physiotherapy and Rehabilitation Sciences, School of Health Sciences, University of Nottingham, Nottingham, UK
[4]Department of Health Professions, Manchester Metropolitan University, Manchester, UK
[5]Research Unit for General Practice in Aalborg, Department of Clinical Medicine at Aalborg University, Aalborg, Denmark
[6]Department of Occupational Therapy and Physiotherapy, Department of Clinical Medicine, Aalborg University Hospital, Aalborg, Denmark
[7]Nuffield Department of Orthopaedics, Rheumatology and Musculoskeletal Sciences, University of Oxford, Oxford, UK

**Twitter** @benedsmith

**Acknowledgements** The authors would like to thank Katie Smith, University Hospitals of Derby and Burton NHS Foundation Trust, who kindly helped with data collection.

**Contributors** BES was responsible for conception and design, compiling the interview schedule, interviewing, transcribing, coding, analysis and interpretation, drafting and revising the manuscript. FM was responsible for conception and design, compiling the interview schedule, coding, analysis and interpretation, drafting and revising the manuscript. PH, MB, JS, MSR, TOS and PL were involved in conception and design, interpretation and reviewing revisions to the manuscript. All authors have read and approved of the final manuscript.

**Funding** This report is independent research arising from a Clinical Doctoral Research Fellowship, Benjamin Smith, ICA-CDRF-2015-01-002 supported by the National Institute for Health Research (NIHR) and Health Education England. TOS is supported by funding from NIHR Oxford Health Biomedical Research Centre.

**Disclaimer** The views expressed in this publication are those of the author(s) and not necessarily those of the NHS, the NIHR, HEE or the Department of Health.

**Competing interests** None declared.

**Patient consent for publication** Not required.

**Ethics approval** This study was approved by the West Midlands—Black Country Research Ethics Committee (16/WM/0414) and Sponsored by University Hospitals of Derby and Burton NHS Foundation Trust. IRAS reference 211417.

**Provenance and peer review** Not commissioned; externally peer reviewed.

**Data sharing statement** Further quotations are available from BES at benjamin. smith3@nhs.net. No additional data are available.

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
