## [Reviewer comments · BMJ Open]

ARTICLE DETAILS

TITLE (PROVISIONAL)	Barriers and facilitators of loaded self-managed exercises and physical activity in people with patellofemoral pain: understanding the feasibility of delivering a multi-centred randomised controlled trial – A UK qualitative study
AUTHORS	Smith, Benjamin; Moffatt, Fiona; Hendrick, Paul; Bateman, Marcus; Selfe, James; Rathleff, Michael; Smith, Toby O.; Logan, Phillipa

VERSION 1 - REVIEW

REVIEWER	M van Middelkoop Erasmus MC, The Netherlands
REVIEW RETURNED	18-Jun-2018

GENERAL COMMENTS	General remarks This is an interesting and valuable study. Though, I do have to admit that I do have a lack of knowledge regarding qualitative methods. I therefore have minor comments and would like to encourage the editorial team to let this paper review by a person specialized and experienced in qualitative work and methods. Specific remarks Introduction: Second paragraph includes a part that suits the methods section and not the introduction. Last sentences of third paragraph need more explanation in order to make sure that all readers get the argument. What is the single exercise method and what shows the literature regarding shoulder pain? This is an unclear argument. Discussion: 2nd paragraph: "This topic needs additional emphasis in any future training programme delivered to the physiotherapists." Please explain how in order to give the reader a more concrete idea on this topic. 3rd paragraph: this needs more explanation in the introduction, as stated above. Page 18, 2nd paragraph: cluster randomisation is one way of overcoming this problem. This is a single sentence and needs more explanation. In what context would it overcome difficulties and how and why?
--

	The conclusion is rather lengthy. I would be nice to have a shorter and as a consequence clearer conclusion.
--	--

REVIEWER	Yubo Fan Beihang University, China
REVIEW RETURNED	14-Nov-2018

GENERAL COMMENTS	General This study explored the barriers and facilitators of loaded self-managed exercises and physical activity in people with patellofemoral pain. Patients and physiotherapists receiving and delivering both the intervention and usual physiotherapy were interviewed. The study concluded that patients' improvements in pain and function may have been mediated by greater self-efficacy and locus of control. The topic of the paper is interesting and relevant. Attitude to pain and locus of control play important role in the rehabilitation progress on PFP. However, in my opinion, the classification and mechanism of pain is the more important basis for determining the rehabilitation strategies. In the case of a sample size of only 10, since PFP is an multifactor syndrome, the difference of exact pathologies of PFP patients may influence the reliability of the study. Specific There are some suggestions to be considered in revising the manuscript.  1. The present methodology was universal for varying exercise therapy, is there any innovation for PFP? 2. Page 5, line 40. PFP is an multifactor syndrome, different pathologies of PFP may lead to varying rehabilitation strategies, which may be different from the conditions of Ref 7 and 8. Thus the reliable of the patient sample size could be questionable. 3. Page 5, line 20 – 47. What's the exact filter rule of the participants, how to manifest their representativeness? 4. Whether the results of the two researchers are obtained independently? What's the difference between their results? 5. Did the author consider the effect of the subject's educational level on pain understanding? 6. Did any statistical method been applied in this study? 7. Besides the patients' belief and locus of control, I think the cause of the pain is an important basis for determining the progress of exercise training. Some minor tolerable pain may be a sign of tissue damage, while some large pains may not affect the healing of the joint.
--

REVIEWER	Rachael Goberman-Hill University of Bristol, UK
REVIEW RETURNED	30-Nov-2018

GENERAL COMMENTS	Thank you for the chance to read this manuscript reporting on the findings of qualitative research within a feasibility RCT. Qualitative research in trials is an important part of feasibility assessment, enabling aspects of trial and intervention design to be explored. This study aims to explore views about a loaded-exercise intervention for people with patellofemoral pain to identify barriers and facilitators to its uptake and delivery. The aim of the study was good and embedding qualitative research in a trial is excellent. The use of Framework is appropriate. Unfortunately I was not convinced about the depth of the data analysed, nor the sampling adequacy. I appreciate that the planned length of interviews was short and the planned sample size had always been 8-10 in each group, but the length of the interviews is remarkably short (mean 11 minutes). The sample size is particularly worrying because in three of the five themes, the analysis compares patients in the intervention group with those in the usual care group as well as comparing physios who did and did not deliver the trial intervention. The sample size and duration of the interviews mean that the conclusions drawn from the comparison are hard to see as transferable, which is a shame. Also, theme five describes physios' development and it doesn't sit easily alongside the other four themes that focus on beliefs and experience of intervention. I could not see that participants had provided their written informed consent to interview, including to be audio-recorded and to the publication of anonymised quotations. I see that they would have consented in writing to the trial but it seems that only verbal consent to audio-recording was given. I normally hope to see written consent to audio-recording and to publication of anonymised quotations and I am sorry if I have missed this. Some sections have a clearer writing style than others, which would have benefited from style editing, for instance 'described narratives' is an unusual phrasing and the description of the epistemology could be much clearer. I do though want to offer some words of encouragement: the authors' use of theoretical insight from behavioural science is welcome and appropriate. The use of qualitative work to explore barrier and facilitators to intervention use and delivery is also welcome.
--

REVIEWER	Lori Bolgla, PT, PhD, MAcc, ATC Augusta University USA
REVIEW RETURNED	09-Dec-2018

GENERAL COMMENTS	Introduction o It would be helpful to briefly explain what is meant by "loaded self-managed exercise protocol" and what encompassed the usual physical therapy intervention (instead of just referencing another article).
--

	 o The introduction appears short. It does not provide a strong justification for having conducted the study. o It would be helpful if the researchers incorporated a theoretical model for comparing the 2 treatment approaches. o Facilitators and barriers to implementation. should be clearly defined and relate back to the reason for the choice of treatment strategies to compare. Results  o I assume that the researchers have pain information for each subject. A recommendation is to add this data to Table 1 to provide the reader not only symptom duration but pain level. This information will provide the readership an idea of tissue irritability. o Experience of the physical therapists spanned a long range, with each using a certain treatment approach for usual treatment. As mentioned above, it will be helpful to have an idea of the interventions used as well as more detail about the self-managed approach. Discussion  o An overall limitation of the discussion relates to the introduction and lack of intervention detail. Having a theoretical model would help explain the results and how they support or did not support the theoretical model. o The discussion appeared to only address locus of control. Could the discussion be expanded to address the importance of the other themes and their impact on rehabilitation? o The clinical implications section could be developed further. As a clinician, it does not provide much guidance of how to implement study findings into clinical practice.
--	--

VERSION 1 – AUTHOR RESPONSE

Reviewer: 1

Reviewer Name: M van Middelkoop

Institution and Country: Erasmus MC, The Netherlands

Please state any competing interests or state 'None declared': None declared

Please leave your comments for the authors below

General remarks

This is an interesting and valuable study. Though, I do have to admit that I do have a lack of knowledge regarding qualitative methods. I therefore have minor comments and would like to encourage the editorial team to let this paper review by a person specialized and experienced in qualitative work and methods.

Thank you for your kind comments.

Specific remarks

Introduction:

Second paragraph includes a part that suits the methods section and not the introduction.

This, and another sentence, has now been moved to the method section.

Last sentences of third paragraph need more explanation in order to make sure that all readers get the argument. What is the single exercise method and what shows the literature regarding shoulder pain? This is an unclear argument.

We have now expanded this paragraph to improve clarity: "The loaded self-managed exercise programme does not align with current UK physiotherapists' preferred treatment approach for PFP."

We have added further intervention descriptions in the paragraph before this: "The loaded self-managed exercise programme is a novel intervention based on pain science (where a single exercise is designed to load and temporarily aggravate patients' symptoms), self-management strategies and improvements in physical activity levels. Usual physiotherapy can be described as a mixed packaged (multi-model) approach of 'trial-and-error' exercises, patellar taping and bracing, and foot orthoses. It is typically aimed at reducing the load on the patella, with avoidance of painful exercise."

Discussion:

2nd paragraph: "This topic needs additional emphasis in any future training programme delivered to the physiotherapists." Please explain how in order to give the reader a more concrete idea on this topic.

This now reads: "This topic needs additional emphasis in any future training programme delivered to the physiotherapists, for example through a dedicated objective in the training package, or via case-study workshops."

3rd paragraph: this needs more explanation in the introduction, as stated above.

This has been addressed above.

Page 18, 2nd paragraph: cluster randomisation is one way of overcoming this problem. This is a single sentence and needs more explanation. In what context would it overcome difficulties and how and why?

To further expand, the sentence now reads: 'Cluster randomisation, where intervention and control participants are located at different recruitment sites, is one way of overcoming what is referred to as "contamination"'

The conclusion is rather lengthy. I would be nice to have a shorter and as a consequence clearer conclusion.

We have trimmed the conclusion to improve length and clarity. The manuscript word count has reduced from 6,624 to 6,530.

Reviewer: 2

Reviewer Name: Yubo Fan

Institution and Country: Beihang University, China

Please state any competing interests or state 'None declared': None declared

Please leave your comments for the authors below

General

This study explored the barriers and facilitators of loaded self-managed exercises and physical activity in people with patellofemoral pain. Patients and physiotherapists receiving and delivering both the intervention and usual physiotherapy were interviewed. The study concluded that patients' improvements in pain and function may have been mediated by greater self-efficacy and locus of control.

The topic of the paper is interesting and relevant. Attitude to pain and locus of control play important role in the rehabilitation progress on PFP. However, in my opinion, the classification and mechanism of pain is the more important basis for determining the rehabilitation strategies. In the case of a sample size of only 10, since PFP is an multifactor syndrome, the difference of exact pathologies of PFP patients may influence the reliability of the study.

Thank you for your comments. The inclusion/exclusion criteria of the patients in the trial was consistent with the current international consensus on PFP classifications.

Crossley et al. 2016 Patellofemoral pain consensus statement from the 4th International Patellofemoral Pain Research Retreat, Manchester. Part 1: Terminology, definitions, clinical examination, natural history, patellofemoral osteoarthritis and patient-reported outcome m. Br J Sports Med 2016;50:839–43.

A sample size of 10 is consistent with the qualitative research methodology paradigm.

Specific

There are some suggestions to be considered in revising the manuscript.

1. The present methodology was universal for varying exercise therapy, is there any innovation for PFP?

As stated in the introduction and method sections, this research was undertaken within a framework of mixed-methods, embedded within a feasibility study looking specifically at an intervention for people with PFP. We have added a brief description of the novel intervention: "The loaded self-managed exercise programme is a novel intervention based on pain science (where a single exercise is designed to load and temporarily aggravate patients' symptoms), self-management strategies and improvements in physical activity levels."

2. Page 5, line 40. PFP is an multifactor syndrome, different pathologies of PFP may lead to varying rehabilitation strategies, which may be different from the conditions of Ref 7 and 8. Thus the reliability of the patient sample size could be questionable.

The inclusion/exclusion criteria of the patients into the trial was consistent with the current international consensus on PFP classifications.

Crossley et al. 2016 Patellofemoral pain consensus statement from the 4th International Patellofemoral Pain Research Retreat, Manchester. Part 1: Terminology, definitions, clinical examination, natural history, patellofemoral osteoarthritis and patient-reported outcome m. Br J Sports Med 2016;50:839–43.

3. Page 5, line 20 – 47. What's the exact filter rule of the participants, how to manifest their representativeness?

We have now added the following sentence in the Participants section; "International consensus has defined PFP symptoms as typically developing insidiously with retropatellar pain or diffuse peripatellar

pain, aggravated by activities that “load the joint”, such as climbing and descending stairs, squatting, running or jumping.”

4. Whether the results of the two researchers are obtained independently? What’s the difference between their results?

These independent notes were not kept, but developed into working analytical framework. As described in the Data Analysis section.

5. Did the author consider the effect of the subject's educational level on pain understanding?

This was not included, but is a good consideration for future research.

6. Did any statistical method been applied in this study?

No statistical methods were applied – the study was conducted within the qualitative research methodology paradigm.

7. Besides the patients’ belief and locus of control, I think the cause of the pain is an important basis for determining the progress of exercise training. Some minor tolerable pain may be a sign of tissue damage, while some large pains may not affect the healing of the joint.

The inclusion/exclusion criteria of the patients into the trial were consistent with the current international consensus on PFP classifications.

A 2016 cross-sectional, case-control study of 64 patients with PFP and 70 pain-free controls demonstrated that structural abnormalities of the PFJ on MRI were not associated with PFP, and therefore is not compatible with finding the ‘cause of the pain’.

van der Heijden RA, Oei EHG, Bron EE, van Tiel J, van Veldhoven PLJ, Klein S, et al. No Difference on Quantitative Magnetic Resonance Imaging in Patellofemoral Cartilage Composition Between Patients With Patellofemoral Pain and Healthy Controls. *Am J Sports Med* 2016;44:1172–8. doi:10.1177/0363546516632507.

van der Heijden RA, de Kanter JLM, Bierma-Zeinstra SMA, Verhaar JAN, van Veldhoven PLJ, Krestin GP, et al. Structural Abnormalities on Magnetic Resonance Imaging in Patients With Patellofemoral Pain: A Cross-sectional Case-Control Study. *Am J Sports Med* 2016;44:2339–46. doi:10.1177/0363546516646107.

Reviewer: 3

Reviewer Name: Rachael Goberman-Hill

Institution and Country: University of Bristol, UK

Please state any competing interests or state ‘None declared’: None known

Please leave your comments for the authors below

Thank you for the chance to read this manuscript reporting on the findings of qualitative research within a feasibility RCT. Qualitative research in trials is an important part of feasibility assessment, enabling aspects of trial and intervention design to be explored. This study aims to explore views about a loaded-exercise intervention for people with patellofemoral pain to identify barriers and facilitators to its uptake and delivery.

The aim of the study was good and embedding qualitative research in a trial is excellent. The use of Framework is appropriate. Unfortunately I was not convinced about the depth of the data analysed, nor the sampling adequacy. I appreciate that the planned length of interviews was short and the planned sample size had always been 8-10 in each group, but the length of the interviews is remarkably short (mean 11 minutes). The sample size is particularly worrying because in three of the five themes, the analysis compares patients in the intervention group with those in the usual care group as well as comparing physios who did and did not deliver the trial intervention. The sample size and duration of the interviews mean that the conclusions drawn from the comparison are hard to see as transferable, which is a shame.

Thank you for your comments and feedback. We feel the sample size was adequate for the epistemology position taken, the qualitative research methodology paradigm presented; the aims of the study, and the conclusions drawn.

Interviewees were targeted based on having specific characteristics, and each one having a unique situation; for example, responding positively or negatively to either interventions or control, returning paper work or not, and gender - with permutations of those variables. The aim of the research was to understand respondents' dynamic qualities of their perceptions and feelings of exercise and pain, and the research design parameters, with qualitative research working towards hypothesis generations, rather than any claims of causation or transferability.

We have added the following sentence in the introduction section setting making clearer the aim of hypothesis generation: "Therefore, the aim of this qualitative investigation was to explore potential barriers and facilitators to implementation of the intervention with participants with PFP involved in a feasibility randomised controlled trial (RCT), with acknowledgment that qualitative inquiry can provide an insight that may lead to development of ideas and hypothesis generation."

We have also added the following sentence in the limitations section of the discussion: "It is possible that the patient sample may differ from other samples within the UK, and how representative these findings are to the greater population of individuals with PFP is unknown."

The conclusions drawn from this research may be very important in intervention development and future efficacy and effectiveness testing of the hypothesis generated. A small sample size doesn't negate the hypothesis generated, and considering the high disability levels, and poor long-term prognosis of young people with PFP, future lines of scientific enquiry are desperately needed.

Additionally, the sample size and mean interview duration is comparable to previous physiotherapy implementation qualitative research, where Littlewood et al's mean duration was 12 minutes.

Littlewood, C., Mawson, S., May, S. and Walters, S., 2015. Understanding the barriers and enablers to implementation of a self-managed exercise intervention: a qualitative study. *Physiotherapy*, 101(3), pp.279-285.

Also, theme five describes physios' development and it doesn't sit easily alongside the other four themes that focus on beliefs and experience of intervention.

Physiotherapists' professional development, personal preference and past experience are thought to act as barriers or facilitators to implementation of research. Therefore, fits tightly within the main aims of the study.

I could not see that participants had provided their written informed consent to interview, including to be audio-recorded and to the publication of anonymised quotations. I see that they would have consented in writing to the trial but it seems that only verbal consent to audio-recording was given. I normally hope to see written consent to audio-recording and to publication of anonymised quotations and I am sorry if I have missed this.

The following sentence is within the participants section: “Initial recruitment to the feasibility study included gaining consent for taking part in future qualitative investigations.”

We have now made the sentence clearer: “Initial recruitment to the feasibility study included gaining written consent for taking part in future qualitative investigations with consent to audio-recording and to publication of anonymised quotations.”

Some sections have a clearer writing style than others, which would have benefited from style editing, for instance ‘described narratives’ is an unusual phrasing and the description of the epistemology could be much clearer.

We have re-written the epistemology description to improve clarity:

“This study did not set out to prove or disprove a hypothesis, it set out to generate new data from which an understanding of barriers and facilitators to the intervention and study design might be developed. The authors took an epistemological position described as “contextualist” by Braun and Clarke that sits central on the spectrum of realism and constructivism. It recognises the experience at an individual level, whilst considering the wider context within a sociocultural perspective. Through this, the beliefs and perceptions of a person, with any meanings attached, can be explored, whilst considering social and cultural factors. This position has previously been discussed in detail in relation to this mixed-methods study.”

I do though want to offer some words of encouragement: the authors’ use of theoretical insight from behavioural science is welcome and appropriate. The use of qualitative work to explore barrier and facilitators to intervention use and delivery is also welcome.

Reviewer: 4

Reviewer Name: Lori Bolgla, PT, PhD, MAcc, ATC

Institution and Country: Augusta University, USA

Please state any competing interests or state ‘None declared’: None

Introduction

o It would be helpful to briefly explain what is meant by “loaded self-managed exercise protocol” and what encompassed the usual physical therapy intervention (instead of just referencing another article).

We have now added the following sentence in the introduction section: “The loaded self-managed exercise programme is a novel intervention based on pain science (where a single exercise is designed to load and temporarily aggravate patients’ symptoms), self-management strategies and improvements in physical activity levels. Usual physiotherapy can be described as a mixed packaged (multi-model) approach of ‘trial-and-error’ exercises, patellar taping and bracing, and foot orthoses. It is typically aimed at reducing the load on the patella, with avoidance of painful exercise.”

o The introduction appears short. It does not provide a strong justification for having conducted the study.

As above, the introduction section has been improved.

o It would be helpful if the researchers incorporated a theoretical model for comparing the 2 treatment approaches.

We have now added the following sentence in the introduction: “The loaded self-managed exercise programme is a novel intervention based on pain science (where a single exercise is designed to load and temporarily aggravate patients’ symptoms), self-management strategies and improvements in physical activity levels. Usual physiotherapy can be described as a mixed packaged (multi-model) approach of ‘trial-and-error’ exercises, patellar taping and bracing, and foot orthoses. It is typically aimed at reducing the load on the patella, with avoidance of painful exercise.”

o Facilitators and barriers to implementation. should be clearly defined and relate back to the reason for the choice of treatment strategies to compare.

Facilitators and barriers are a well-recognised scientific term, and do not relate specifically to treatment strategies, but to the adherence of the treatment strategy.

Results

o I assume that the researchers have pain information for each subject. A recommendation is to add this data to Table 1 to provide the reader not only symptom duration but pain level.

This information will provide the readership an idea of tissue irritability.

This would be in breach of the protocol, and therefore not possible.

o Experience of the physical therapists spanned a long range, with each using a certain treatment approach for usual treatment. As mentioned above, it will be helpful to have an idea of the interventions used as well as more detail about the self-managed approach.

We have now added the following sentence in the introduction: “The loaded self-managed exercise programme is a novel intervention based on pain science (where a single exercise is designed to load and temporarily aggravate patients’ symptoms), self-management strategies and improvements in physical activity levels. Usual physiotherapy can be described as a mixed packaged (multi-model) approach of ‘trial-and-error’ exercises, patellar taping and bracing, and foot orthoses. It is typically aimed at reducing the load on the patella, with avoidance of painful exercise.”

Discussion

o An overall limitation of the discussion relates to the introduction and lack of intervention detail. Having a theoretical model would help explain the results and how they support or did not support the theoretical model.

We have now added further details on the two treatment approaches in the introduction.

o The discussion appeared to only address locus of control. Could the discussion be expanded to address the importance of the other themes and their impact on rehabilitation?

This is not correct, the discussion included: main findings; dealing with flare-ups; single exercise approach; locus of control; department culture; fidelity; physiotherapists’ past belief and expectations’;

and study limitations and strength. Locus of control is one paragraph out of ten. Although, locus of control was one overarching theme that was evident throughout all themes.

o The clinical implications section could be developed further. As a clinician, it does not provide much guidance of how to implement study findings into clinical practice.

This study is about research design implementation and hypothesis generation. Findings shouldn't be implemented into clinical practice.

VERSION 2 – REVIEW

REVIEWER	Yubo Fan Beihang University, China
REVIEW RETURNED	21-Feb-2019

GENERAL COMMENTS	General Authors have replied the prior questions and supplemented the information accordingly. There are still some suggestions for this article. Specific Although the inclusion/exclusion criteria of the patients into the trial was consistent with the current international consensus on PFP classifications, the pathologies of PFP is still varying within the inclusion criteria. For example, from the perspective of orthopedics, pain in different activities (e.g. ascending stairs, descending stairs, and running) may mean different types and locations of tissue damage. Ignoring the specific mechanisms of each PFP case will limit the effectiveness of rehabilitation strategies. And the individual differences will affect the conclusions of the study in the case of small sample sizes.
--

VERSION 2 – AUTHOR RESPONSE

Reviewer(s)' Comments to Author:

Reviewer: 2

Reviewer Name: Yubo Fan

Institution and Country: Beihang University, China

Please state any competing interests or state 'None declared': None declared

Please leave your comments for the authors below

General

Authors have replied the prior questions and supplemented the information accordingly. There are still some suggestions for this article.

Specific

Although the inclusion/exclusion criteria of the patients into the trial was consistent with the current international consensus on PFP classifications, the pathologies of PFP is still varying within the inclusion criteria. For example, from the perspective of orthopedics, pain in different activities (e.g. ascending stairs, descending stairs, and running) may mean different types and locations of tissue damage. Ignoring the specific mechanisms of each PFP case will limit the effectiveness of rehabilitation strategies. And the individual differences will affect the conclusions of the study in the case of small sample sizes.

We have acknowledged that our patient sample may differ from other sampled within the UK. Please see last paragraph of study limitations and strengths section: “It is possible that the patient sample may differ from other samples within the UK, and how representative these findings are to other populations with PFP is unknown.”

However, we disagree with your assertion that pain on different activities may mean different types of locations of tissue damage. Traditional pain models that describe tissue pathology as a source of nociceptive input directly linked with pain expression are insufficient for assessing and treating musculoskeletal pain. Other models reconceptualise pain and put forward concepts that are based on the premise that pain does not always provide a measure of the state of tissue pathology. Instead, pain is modulated by many factors, and the relationship between pain and tissue becomes less predictable the longer pain persists.

- Smith BE, Hendrick P, Bateman M, et al Musculoskeletal pain and exercise—challenging existing paradigms and introducing new Br J Sports Med Published Online First: 20 June 2018. doi: 10.1136/bjsports-2017-098983
- Smith BE, Hendrick P, Smith TO, et al Should exercises be painful in the management of chronic musculoskeletal pain? A systematic review and meta-analysis Br J Sports Med 2017;51:1679-1687.
- Moseley, G.L., 2007. Reconceptualising pain according to modern pain science. Physical therapy reviews, 12(3), pp.169-178.

We have therefore not made any further changes to the manuscript.